# A Biochemical Platform to Define the Relative Specific Activity of *IDUA* Variants Identified by Newborn Screening

**DOI:** 10.3390/ijns6040088

**Published:** 2020-11-12

**Authors:** Seok-Ho Yu, Laura Pollard, Tim Wood, Heather Flanagan-Steet, Richard Steet

**Affiliations:** Greenwood Genetic Center, Greenwood, SC 29646, USA; syu@ggc.org (S.-H.Y.); lpollard@ggc.org (L.P.); tim@ggc.org (T.W.); heatherfs@ggc.org (H.F.-S.)

**Keywords:** *IDUA*, α-iduronidase, newborn screening, pseudodeficiency, mucopolysaccharidosis

## Abstract

The lysosomal storage disorder, mucopolysaccharidosis I (MPSI), results from mutations in *IDUA*, the gene that encodes the glycosaminoglycan-degrading enzyme α-L-iduronidase. Newborn screening efforts for MPSI have greatly increased the number of novel *IDUA* variants identified, but with insufficient experimental evidence regarding their pathogenicity, many of these variants remain classified as variants of uncertain significance (VUS). Defining pathogenicity for novel *IDUA* variants is critical for decisions regarding medical management and early intervention. Here, we describe a biochemical platform for the characterization of *IDUA* variants that relies on viral delivery of *IDUA* DNA into *IDUA*-deficient HAP1 cells and isolation of single cell expression clones. The relative specific activity of wild-type and variant α-iduronidase was determined using a combination of Western blot analysis and α-iduronidase activity assays. The specific activity of each variant enzyme was consistent across different single cell clones despite variable *IDUA* expression and could be accurately determined down to 0.05–0.01% of WT α-iduronidase activity. With this strategy we compared the specific activities of known pseudodeficiency variants (p.His82Gln, p.Ala79Thr, p.Val322Glu, p.Asp223Asn) or pathogenic variants (p.Ser633Leu, p.His240Arg) with variants of uncertain significance (p.Ser586Phe, p.Ile272Leu). The p.Ser633Leu and p.His240Arg variants both show very low activities consistent with their association with Scheie syndrome. In our experiments, however, p.His240Arg exhibited a specific activity five times higher than p.Ser633Leu in contrast to other reports showing equivalent activity. Cell clones expressing the p.Ser586Phe and p.Ile272Leu variants had specific activities in the range of other pseudodeficiency variants tested. Our findings show that pseudodeficiency and pathogenic variants can be distinguished from each other with regard to specific activity, and confirms that all the pseudodeficiency variants variably reduce α-iduronidase activity. We envision this platform will be a valuable resource for the rigorous assessment of the novel *IDUA* variants emerging from the expansion of newborn screening efforts.

## 1. Introduction

The laboratory diagnosis of MPSI (Hurler syndrome) historically began with the observation of elevated urinary GAGs, specifically dermatan and heparan sulfate. Following a positive urinary screening test, blood was collected for measurement of α-iduronidase activity. Activity measurements below a certain threshold were considered the gold standard for diagnosis. In regions where DNA-based analysis was available *IDUA* gene sequencing would follow. This set of three laboratory tests, urinary GAG, functional enzyme analysis and gene sequencing, allowed the physician to confidently provide the diagnosis of MPSI for a child with expected clinical features [1]. With the advent of treatments for MPSI, and the realization that early intervention provides the highest benefit, a timely and accurate diagnosis of MPSI has become essential. Newborn screening for MPSI began in Missouri in 2013 and continues to expand to other states [2,3,4,5]. Because the features of MPSI are not present at birth, clinical phenotypes cannot be leveraged as part of the diagnostic algorithm following NBS. The definitive diagnosis of MPSI, and classification as either the severe Hurler or attenuated Scheie, must be made by laboratory studies alone. The newborn screening program uses α-iduronidase activity measured in dried blood spots with low enzyme levels considered a positive screen. Measurement of enzyme activity serves as the primary standard for the diagnosis of many lysosomal storage conditions. More recently, second tier GAG testing from dried blood spots has been employed as a means of confirming or ruling out an MPSI positive screen [6,7,8]. This testing is advantageous since it does not require *IDUA* sequencing to help address the validity of the NBS activity measurement. Nonetheless, there is still a subset of cases described in two of the papers cited above where the GAG testing does not provide clear support for the enzyme analysis and further molecular testing is necessary [7,8]. Moreover, enzyme activity in some clinically normal individuals can be below the established normal range, and in some cases the reduction in enzyme activity may be similar to what is found in affected patients [9]. This phenomenon, termed pseudodeficiency, has been described for several lysosomal conditions including Pompe disease, metachromatic leukodystrophy and Tay–Sachs disease [10,11,12]. Pseudodeficiency is problematic in the context of a positive newborn screen as it can create increased false positives until the pseudodeficiency alleles in a given population can be identified and characterized. 

In light of the fact that isolating and sequencing DNA from dried blood spots is relatively simple, *IDUA* sequencing became the primary clinical test for physicians following infants with a positive MPSI newborn screen. Sanger sequencing or NGS assays can easily detect most mutations in the 14- exon *IDUA* gene. While the identification of a variant is straightforward, determining the functional significance of rare missense variants and classifying them as either pathogenic or benign can be challenging particularly if it has not been described in the literature or public databases. The American College of Medical Genetics and Genomics (ACMG) have published a set of guidelines for variant classification [13]. Several factors are taken into account, such as in silico predicted impact of the change, literature review, and frequency of the change in public SNP databases, along with other factors. Each factor is scored according to its relative strength of pathogenicity. A tally of the information is made and if there is sufficient information, a variant can be classified as pathogenic or benign. However, many variants lack sufficient evidence and are termed a “variant of uncertain significance” or “VUS”. One criterion that can have a strong impact on the ACMG system is whether functional studies of the individual variant have been performed in in vitro systems. In these laboratory studies, a specific sequence variant is analyzed in isolation and its specific function on enzyme activity and cellular catabolism is investigated.

Unfortunately, the diagnostic laboratories that identify these changes often have little resources or scientific expertise to perform important functional analyses. Therefore, the time between the initial identification of a variant and gathering enough data according to ACMG guidelines to determine if a variant is pathogenic or benign is several years. In a newborn screening setting, caregivers and families are left to take a “wait and see” approach, with serial measurement of urinary GAGs and the lack of clinical features for reassurance. Some predictive strategies have been developed that combine the measurement of enzymatic activity with GAG assays in dried blood spots [14,15,16]. These strategies, however, do not directly address how individual variants impact the α-iduronidase enzyme. Cell-based platforms have also been successfully developed by several groups to gauge α-iduronidase specific activity, with some limitations in the determination of residual activity for certain variants [17,18,19]. To overcome these hurdles, we developed a new biochemical platform that can be used to functionally characterize novel *IDUA* variants identified through newborn screening. We used this platform to more precisely define the specific activity of several known *IDUA* mutations (associated with both severe Hurler and attenuated Scheie), pseudodeficiency variants, and two novel VUS, allowing us to accurately classify their pathogenicity. The inclusion of Scheie-associated variants proved useful to compare to the pseudodeficiency variants. Several new observations regarding the impact of these variants on α-iduronidase processing and activity, as well as the future modification for the broad application of this platform are discussed.

## 2. Materials and Methods

The lentiviral particles containing WT or mutant *IDUA* cDNA were generated by the Viral Vector Core at the University of South Carolina School of Medicine (Columbia, SC, USA). Anti-α-iduronidase antibody (PA5-45483) was purchased from Invitrogen. 4-Methylumbelliferyl-α-iduronic acid (4-MUI; M334700) was purchased from Toronto Research Chemicals. HAP1 WT or *IDUA* KO cell line was purchased from Horizon Inspired Cell Solutions. Hexadimethrine bromide (polybrene, H9268) was obtained from Sigma Aldrich and Protease inhibitor cocktail (88666) was from Thermo Scientific (Waltham, MA, USA).

### 2.1. Transduction of HAP1 IDUA KO Cells and Single Cell Cloning

HAP1 *IDUA* KO cells were cultured using Iscove’s Modification of Dulbecco’s Media (IMDM) with 10% fetal bovine serum (FBS, Benchmark, Sayreville, NJ, USA) and penicillin (100 IU/mL)/streptomycin (100 µg/mL, Media Tech, Manassas, VA, USA) in a 5% CO2 atmosphere, 37 °C humid incubator. Cells were plated at 10% confluency in a 12 well dish 24 h before transduction. Next day, transduction was performed by treating each well of cells with 3 µL or 11 µL of lentivirus (from 8 × 10^8^ TU/mL or 2 × 10^8^ TU/mL stocks, respectively) containing WT or mutant *IDUA* DNA in 1 mL of complete IMDM media containing 6 µg/mL of polybrene. After 24 h, the transduction media was replaced with fresh complete IMDM. After another 24 h, puromycin (1.0 µg/mL) was added to each well for three days, and the dead cells were removed by washing the well with Dulbecco’s Phosphate-Buffered Saline (DPBS). The survived cells were cultured with 0.1 µg/mL of puromycin (lower concentration used for maintenance) and replated in 6 cm dishes when cell number was sufficient. After culturing the cells as a mixed population, single cell cloning was performed by limiting dilution in a 96 well plate to isolate stably transduced cell clones that overexpress *IDUA*. Verification of stably transduced cell clones was performed by measuring α-iduronidase activity, or in some cases by Western blot using the anti-α-iduronidase antibody.

### 2.2. α-Iduronidase Activity Assay

Typically, cultured cells were harvested by trypsinization and cell pellets lysed by sonication with DPBS containing Ca^2+^, Mg^2+^, protease inhibitor cocktail, and 0.1% Triton X-100, followed by incubation on ice for 30 min. After centrifugation at 20,000× *g* for 10 min, the resulting supernatant was saved and protein concentration was determined by BCA assay kit (Thermo Scientific, Waltham, MA, USA). Protein concentration of each lysate was typically adjusted to 0.1 or 0.01 mg/mL. 4-methylumbellyferyl-α-d-iduronic acid (10 mM in DMSO) was diluted to 100 µM with 0.4 M NaOAc buffer (pH 3.5). Enzyme reactions were initiated by mixing each lysate (25 µL) with 4-methylumbellyferyl-α-d-iduronic acid (25 µL) in a 96 well black plate followed by incubation at 37 °C for 1 h. Reactions were terminated with the addition of 200 µL of 0.1 M glycine buffer (pH 10.4) and fluorescence was measured (excitation/emission = 355 nm/460 nm) with Cytation™ 5 Cell Imaging Multi-Mode Reader (BioTek, Winooski, VT, USA). To obtain relative activity of mutant α-iduronidase (with the same protein concentration of lysate) compared to WT α-iduronidase, fluorescence from WT *IDUA-*expressing cells was measured in serially diluted lysates (for example, 0.01 mg/mL–0.00001 mg/mL) to produce a standard curve. Fluorescence of mutant α-iduronidase in cell lysates was determined using the calibration curve to determine a matching concentration of WT *IDUA-*expressing cell lysate that gives the equivalent fluorescence. Then, the ratio of the concentrations of mutant α-iduronidase/WT α-iduronidase activity was used to establish relative enzyme activity. 

### 2.3. Immunoblot and Determination of Relative Expression Level of IDUA

The same lysates used for the α-iduronidase activity assay (typically at 2.0 mg/mL) were resolved by SDS-PAGE and transferred to nitrocellulose membranes. Immunoblotting was done with anti-α-iduronidase antibody (1:500, overnight) followed by incubation with anti-rabbit HRP-conjugated secondary antibody (1:2000, 1 h). The ChemiDoc™ imaging System (BioRad, Hercules, CA, USA) was used to obtain Western blots and to analyze the relative amount of α-iduronidase expressed in each transduce cell clones. In order to more accurately obtain the relative α-iduronidase abundance in each of the clones tested, we loaded a range of different protein amounts (40, 20, 10, and 5 µg) from one of our WT clones (WT-1B3). The same amount of total lysate protein was resolved in each lane of this standard range by adding the necessary amount of IDUA KO lysate to reach 40 µg. Normalization of this relative enzyme activity with the amount of α-iduronidase signal measured by densitometry of Western blots allowed the determination of relative specific activity for each variant-bearing form of α-iduronidase. For loading control, Ponceau S staining on the nitrocellulose membrane was performed.

## 3. Results

To create a robust cell-based platform to assess *IDUA* variant pathogenicity, individual cell lines expressing either WT IDUA or one of several variant-containing sequences were generated. The overall workflow of this platform is depicted in Figure 1. To establish a reliable scale for variant analyses, α-iduronidase activity was first compared in WT and *IDUA*-deficient HAP1 cells, as well as deficient cells transfected with WT DNA (Figure 2A). Since HAP1 cells have very low levels of endogenous α-iduronidase, activity was only slightly higher in parental control cells than that noted in *IDUA*-deficient cells. Similarly, α-iduronidase was undetectable by Western blot in both the parental and *IDUA-*deficient line (not shown). Transfection with WT IDUA DNA increased activity five times that of the untransfected control. However, since transfection efficiency is notoriously low in HAP1 cells we instead utilized a lentiviral expression system to create cell lines that stably express α-iduronidase. For this enzyme deficient HAP1 cells were incubated with *IDUA*-expressing lentiviral particles and transduced cells enriched with puromycin. Following limiting dilution at least two different single cell clones were isolated for each *IDUA* variant and analyzed for α-iduronidase activity, protein abundance, and processing.

The platform was established and validated by comparing WT *IDUA*, a catalytically dead variant (p.Glu182Lys), and a small set of sequence variants (Table 1, see refs in Table 1). Analyses included several known pseudodeficient variants (p.Ala79Thr, p.His82Gln, p.Asp223Asn, p.Val322Glu), a Scheie-associated variant (p.Ser633Leu) and two different variants of uncertain significance (p.Ile272Leu and p.Ser586Phe). The 48 h post-infection mixed cell populations transduced with WT *IDUA* exhibited 19 times more α-iduronidase activity than the parental control line (Figure 2B), which increased to ~530 times more α-iduronidase activity following puromycin selection. The activity levels of individual clonal cell lines expressing WT enzyme (i.e., WT-1B3, 1C8, 1C10) ranged from 3500 to 5800 times that of the control (Figure 2B). Using the WT-1B3 line, a standard curve was created by calculating the percent of fluorescence-based activity relative to the concentration of lysate (mg/mL) in the reaction (Appendix A). To maintain linearity within the assay, measurements were made using lower concentrations. The standard curve was created such that the calculated ratio could be used to assess the relative activity (calculated as a percent of the control reference, WT-1B3) for each *IDUA* variant (Figure 3A).

To more accurately assign a variant’s “specific” activity relative to its own expression, Western blot was used to quantitate α-iduronidase levels by pixel densitometry (Figure 3B). Unlike formal specific activity measurements that utilize a known quantity of purified protein in an activity assay, Western blot densitometry provides an estimate of α-iduronidase abundance that can be compared between samples or cells with different levels of enzyme expression (such as those that express the WT α-iduronidase enzyme). This relative specific activity is calculated by dividing each α-iduronidase variant’s relative activity by its relative abundance (Figure 3B,C). For this, protein abundance relative to the WT control was determined by generating a standard curve with different WT-1B3 lysate concentrations (5, 10, 20, 40 µg). This was an important parameter when comparing the activity of different cell lines, as *IDUA* expression varied among the different alleles tested. Notably, protein levels were substantially reduced (10–30% of WT-1B3) in many cell lines expressing either a pseudodeficiency variant or VUS (Figure 3B). These data suggest that the *IDUA* sequence alterations tested may have an effect on mRNA half-life, translation efficiency and/or enzyme stability. Although reduced enzyme abundance was noted in multiple clones, this may simply reflect the location of viral integration among the different clones.

In addition to differences in the level of α-iduronidase abundance, Western blot analyses revealed several variants with alterations in the extent enzyme processing. Following removal of the signal peptide in the early secretory pathway, the α-iduronidase enzyme is processed into ten different peptides within the endosomal/lysosomal compartment, ranging in size from 74 kDa to 5 kDa. We were able to detect at least of the processed peptides in some of the clones. For example, we detected the 44 kDa peptide for WT α-iduronidase and noted the 69 kDa peptide for the p.Glu182Lys variant enzyme. Interestingly, all other variants tested exhibited decreased abundance of processed peptides. This included the pseudodeficient variants (p.Ala79Thr, p.His82Gln, p.Val322Glu), the Scheie-associated variant (p.Ser633Leu) and the variant of uncertain significance (p.Ser586Phe).

Two *IDUA* sequence variants (p.Glu182Lys and p.Ser633Leu) showed very low activity (0.001–0.1%). To accurately measure this, we performed the activity assay using higher concentrations of lysates (2.0 mg/mL; Appendix A). To account for background fluorescence, lysate from *IDUA*-deficient cells was used as a control. As shown, p.Glu182Lys-C5 or p.Ser633Leu clones clearly showed concentration-dependent increases in fluorescence. The specific activity of the p.Glu182Lys-D11 was close to background (specific relative activity = ~0.001% of WT-1B3), but was reproducibly measured despite being near the limit of detection. Using this method a specific relative activity of (0.03–0.1%) was assigned for the p.Glu182Lys-C5 or p.Ser633Leu *IDUA* variants. In contrast, the p.Ala79Thr, p.His82Gln, and p.Val322Glu *IDUA* clones all exhibited relative specific activities that were at least 5% of WT-1B3 activity, indicative of being pseudodeficient variants. Importantly, this analysis provides direct evidence that these variants do impact α-iduronidase enzyme activity but are not pathogenic for MPSI. Similarly, analysis of the previously designated VUS p.Ser586Phe revealed a relative specific activity of 18%, which also places it in the pseudodeficient range.

Three additional *IDUA* variants (p.Asp223Asn, p.His240Arg, p.Ile272Leu) were investigated to determine their relative specific activity (Figure 4). For comparison, we included WT-1B3 and A79T-F11 in this second set. In case of A79T-F11, specific activity was measured as ~18% vs. the previous run (~8%) with higher fluorescence of the lysate as well. These deviations in the specific α-iduronidase activity could be partially due to 1) variation in the quantification of α-iduronidase by Western blot, or 2) different culture conditions at the time of harvest that could result in altered processing status of α-iduronidase. Nevertheless, WT-1B3 showed similar fluorescence as before, and specific activities of mutant IDUAs were analyzed again as relative values compared to the WT-1B3. p.Asp223Asn variant *IDUA* showed ~13% activity, p.Ile272Leu showed 54–59% activity. The p.His240Arg variant showed a much lower activity (0.4 ~ 0.9% relative to WT-1B3), however, this was ~10 times higher than p.Ser633Leu (0.05 ~ 0.09% activity). As these two variants were previously reported to exhibit similar residual activity [15], we performed our analysis on two more biological replicates of the p.His240Arg and p.Ser633Leu cell clones. The average specific activity from the triplicate analysis of these two variants, and all the other relative specific activities, are shown in Figure 5. Based on this chart, one can interpret variants with relative specific activity that is above the line as pseudodeficient and those below the line as pathogenic, although we acknowledge that the number of variants studied will need to increase before reliable boundaries can be established.

## 4. Discussion

With the number of novel *IDUA* variants identified through newborn screening efforts continuing to increase, the need for a robust biochemical platform that can be used to test variant pathogenicity and confirm pseudodeficiency is critical. The platform described here provides an estimation of the specific activity of variant-bearing α- iduronidase enzymes when expressed in cells lacking endogenous *IDUA*. By normalizing enzyme activity towards a simple sugar substrate to the amount of protein detected by Western blot, a relative specific activity can be determined and used to compare the impact of different variants on enzyme function. A notable strength of this platform is its sensitivity (i.e., the ability to accurately determine relative activity for enzymes with very low residual activity). Leveraging the natural sensitivity of fluorescence measurements with the overexpression of variant enzyme in our cell-based system, we were able to discern variants even with very low residual activity of α-iduronidase (e.g., 0.05–0.1% of WT α-iduronidase). The selection and growth of transduced cells allowed us to isolate multiple single cell clones expressing each variant enzyme. While these clones exhibit variable expression of the α-iduronidase enzyme, the estimated relative specific activity was largely consistent between the clones, supporting the fidelity of this platform.

Specific activity estimations at the lower end of the range are important to distinguish Scheie and Hurler-associated variants, especially when they are combined with other unknown variants. For example, p.Ser633Leu-expressing cell clones showed 0.05 ~ 0.1% relative specific activity. A recent report demonstrated that a patient with homozygous p.Ser633Leu mutation showed an attenuated phenotype despite this low α-iduronidase activity [17]. Individuals with an p.His240Arg *IDUA* variant have also been reported to show an attenuated phenotype when this variant is present in combination with p.Trp402* (compound heterozygous) [18]. Our measurement of p.His240Arg relative specific activity was in the range between 0.4–0.9%. Thus, we might predict a more attenuated phenotype in individuals that are homozygous for the p.His240Arg variant, if there is, than the case of homozygous p.Ser633Leu mutation [17]. Indeed, a patient with p.Ser633Leu/p.Ala75Thr has been reported to have Hurler syndrome [19]. The compound heterozygous mutations (p.Trp402*/Ser633Leu) have been reported as being associated with Hurler and Scheie, while other cases (Pro533Arg/Ser633Leu, Ser633Leu/p.Ser16_Ala19del, Leu409Pro/Ser633Leu) all caused attenuated (Scheie or Hurler/Scheie) disease [17].

The ability to accurately measure the low range activity of variant *IDUA* (0.1–1.0%) with a good signal to noise ratio is important as it may help to better stratify the resulting phenotypic consequences associated with different variants. If the specific activity of a novel *IDUA* variant were compared with the activity of p.Ser633Leu and p.His240Arg *IDUA*, it may be possible to predict where the *IDUA* variant would be associated with the attenuated Scheie or more pronounced Hurler phenotype. Of course, any prediction of phenotypic outcome would need to incorporate the impact of the second allele. Based on p.His240Arg (0.4–0.9%), a threshold residual specific activity seems to be between 2–5% or higher for the pseudodeficiency variants. Our findings demonstrate, however, that these pseudodeficiency variants all do have lower relative specific activity. It is interesting to speculate that such variants share a common mechanism of action that reduce enzymatic activity, although further experimentation is needed to address this question. The p.His82Gln (26–43%), p.Ala79Thr (7–18%), p.Val322Glu (9–11%) and p.Asp223Asn (12–13%) have all been reported as pseudodeficiency variants [20,21]. Previously considered VUS, both p.Ser586Phe (16–17%) and p.Ile272Leu (54–59%) should be reclassified as pseudodeficiency variants. Again, the actual expression level is an important factor to consider and it is possible for patients with same mutation can show different phenotype. We saw relatively different *IDUA* expression level in each of the stable clones by Western blot. Apparently, wild type *IDUA* or Hurler variant (p.Glu182Lys) seems to be expressed more efficiently while other mutants show significantly reduced expression level. Generally speaking, we detected fewer processed forms of α-iduronidase when characterizing the different variant enzymes, and in comparison to prior studies using CHO cells [14,22,23]. There were, however, some consistency between our study and those carried out in CHO cells. For example, both noted lower MW processed forms of α-iduronidase for the p.Glu182Lys variant. Due to the limited recognition of endogenous α-iduronidase with currently available antibodies, direct measurement of α-iduronidase in patient cells has proven more challenging.

While this platform offers an accurate estimation of relative specific activity, it is limited with regard to speed. This is mainly due to the slow production of lentiviral particles and the relatively inefficient transduction of the HAP1 cells. To accelerate variant classification in the future, we are currently exploring the efficiency of transient transfection to deliver variant-bearing DNAs into an *IDUA*-deficient HEK293 cell line (followed by selection for stable transfectants). This approach has the advantage of lower cost and faster turnaround due to the high transfection efficiency of HEK293 cells. As an added level of assessment, we believe the analysis of GAG storage using assays that monitor the abundance of fragments with non-reducing ends in these clonal cell lines would allow us to correlate α-iduronidase specific activity with functional clearance of GAG storage. One significant caveat, however, is the overexpression of *IDUA* in these clonal lines. While this does not appear to influence our specific activity measurements, high-level expression of α-iduronidase with reduced activity might still maintain enough intralysosomal hydrolytic capacity to avoid detectable GAG storage.

## Figures and Tables

**Figure 1 IJNS-06-00088-f001:**
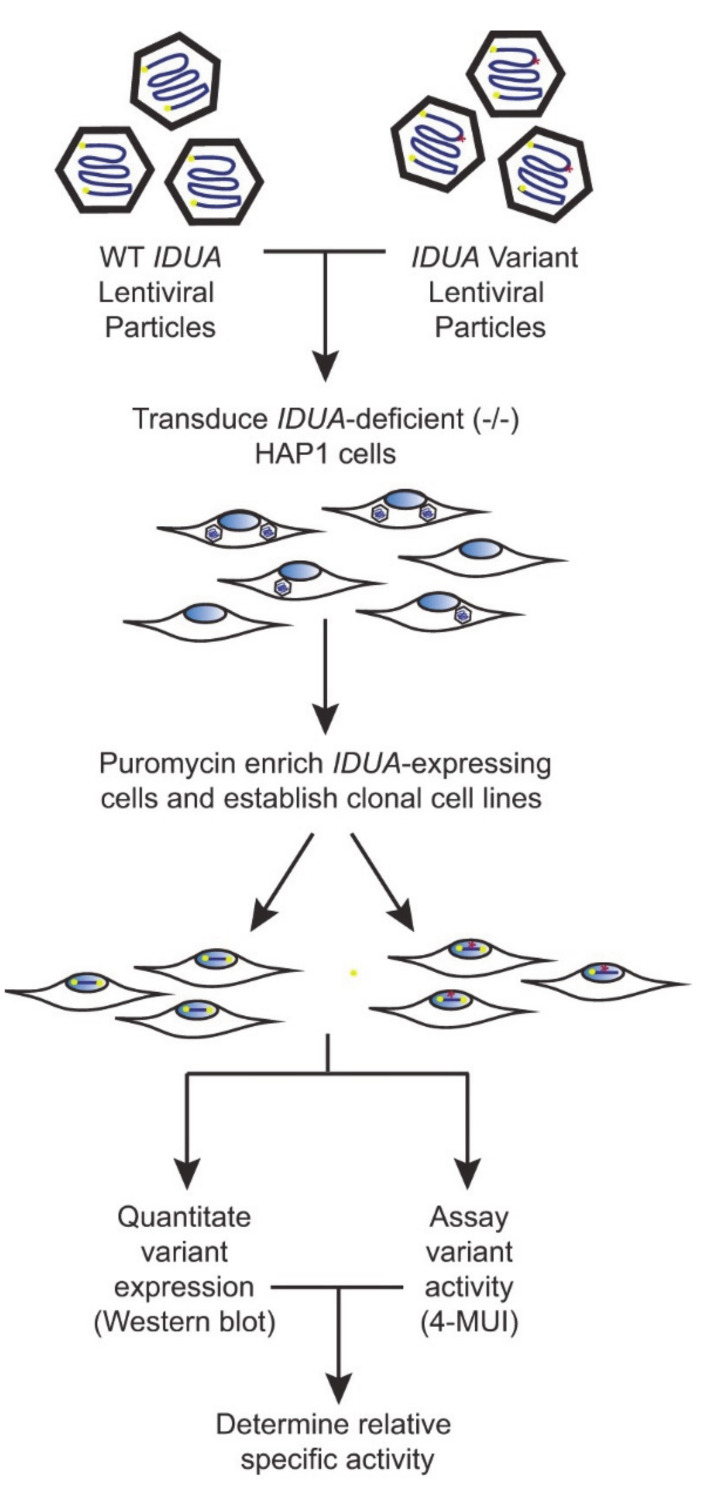
Schematic of the biochemical platform.

**Figure 2 IJNS-06-00088-f002:**
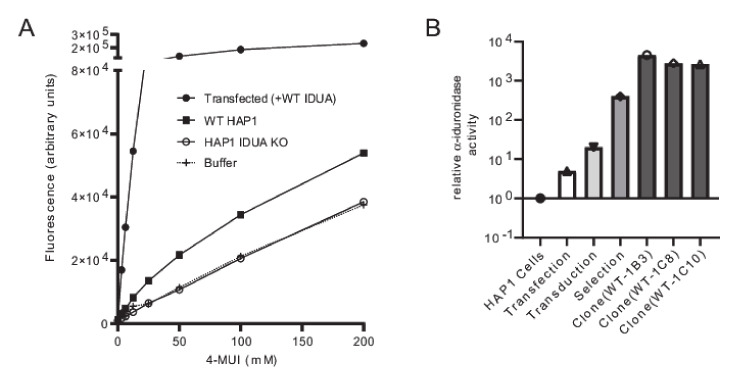
Validation of the *IDUA*-KO HAP1 cells and α-iduronidase activity enrichment during the selection process. (**A**) Enzyme activity in WT and *IDUA*-KO HAP1 cells compared α-iduronidase activity in *IDUA*-KO HAP1 cells transfected with WT *IDUA* cDNA. (**B**) Enrichment of α-iduronidase activity across different steps of the selection process.

**Figure 3 IJNS-06-00088-f003:**
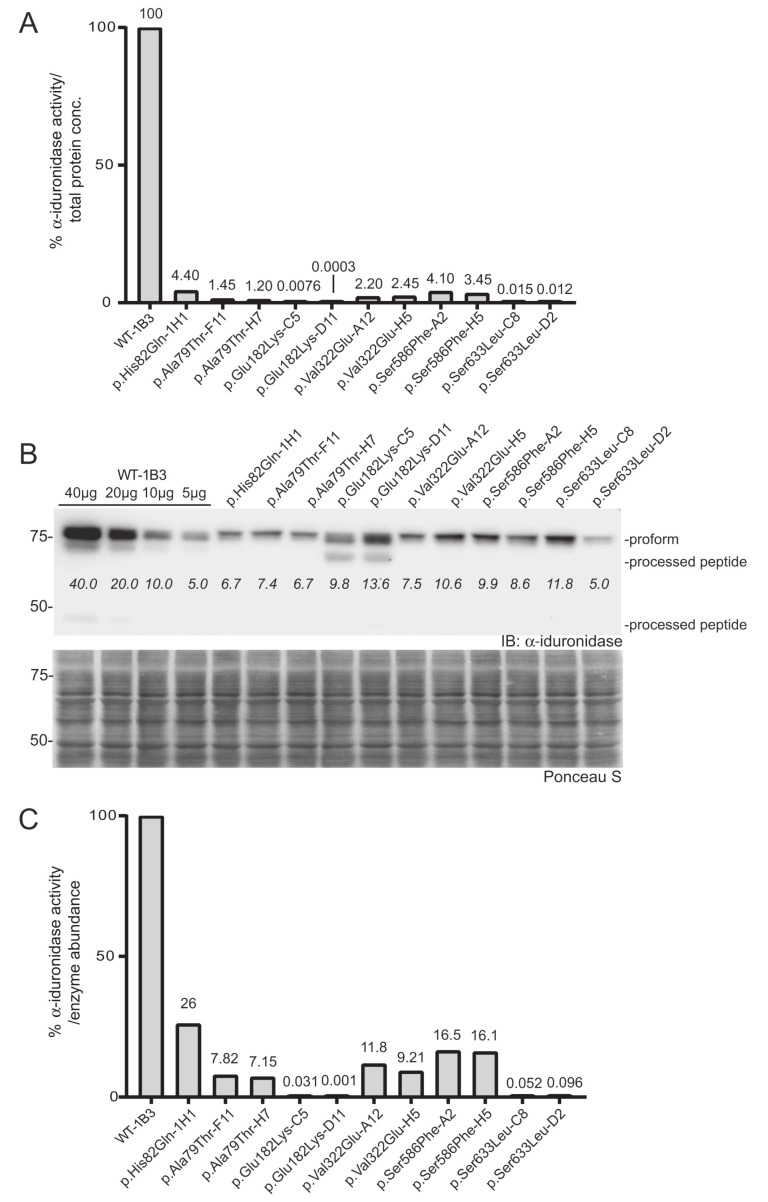
Determination of the relative specific activity in WT and variant *IDUA*- expressing cell clones. (**A**) Relative α-iduronidase activity in cell lysates from different single cell clones. (**B**) Representative Western blot for α-iduronidase in the cell lysates from these clones. Ponceau S staining was used to confirm equivalent total protein loading. (**C**) Relative specific activity determination in the clones calculated by dividing relative enzyme activity by relative α-iduronidase abundance as measured by densitometry.

**Figure 4 IJNS-06-00088-f004:**
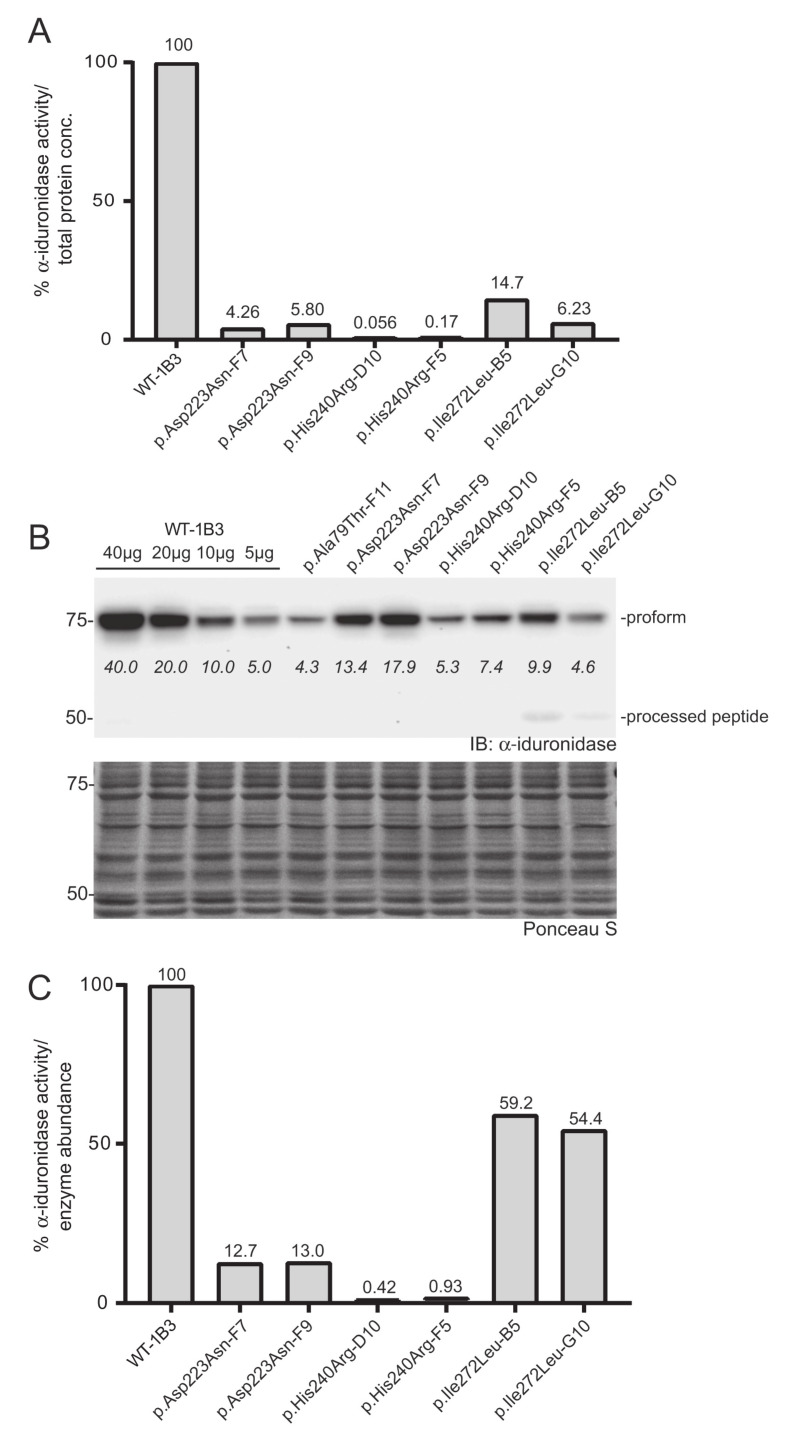
Determination of the relative specific activity in WT and variant *IDUA*-expressing cell clones. (**A**) Relative α-iduronidase activity in cell lysates from different single cell clones. (**B**) Representative Western blot for α-iduronidase in the cell lysates from these clones. Ponceau S staining was used to confirm equivalent total protein loading. (**C**) Relative specific activity determination in the clones calculated by dividing relative enzyme activity by relative α-iduronidase abundance as measured by densitometry.

**Figure 5 IJNS-06-00088-f005:**
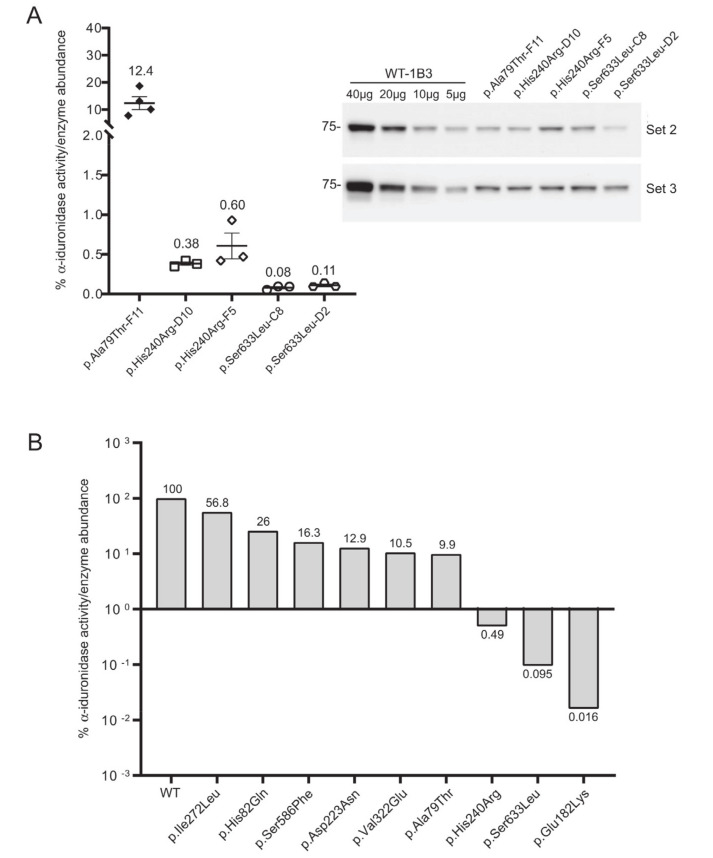
**(A)** Repeat analysis of the p.His240Arg and p.Ser633Leu variants using two different clones for each variant. Average relative specific activity is shown. Error bars represent standard error of the mean. **(B)** Summary of relative specific activity for all variants in this study.

**Table 1 IJNS-06-00088-t001:** List of IDUA variants and their classification within this study.

Variant	Classification	References
p.Ala79Thr	pseudodeficient	[20]
p.His82Gln	pseudodeficient	[20]
p.Glu182Lys	Pathogenic (Hurler)	[21]
p.Asp223Asn	pseudodeficient	[20]
p. His240Arg	Pathogenic (Scheie)	[16,17]
p.Ile272Leu	VUS	N.R.
p.Val322Glu	pseudodeficient	[20]
p.Ser586Phe	VUS	[15]
p.Ser633Leu	Pathogenic (Scheie)	[16,17,18]

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
