# Peer review of "A Biochemical Platform to Define the Relative Specific Activity of *IDUA* Variants Identified by Newborn Screening"

_2409-515X, 2020, doi:10.3390/ijns6040088_

Round 1

Reviewer 1 Report

Journal Review: IJNS IDUA Variants identified by NBS

This manuscript provides a highly sensitive mechanism to classify variants of unknown significance in the IDUA gene that were identified through newborn screening for MPS I. It will be valuable to the newborn screening and medical community in determining the prognosis of newborns who screen positive for MPS I and are found to have a VUS through molecular testing.

This manuscript is well written, and the study is well thought out. I have a few minor points for the authors to consider:

  1. In introduction, line 56, it states that this leaves only urinary GAG testing and IDUA gene studies. This is no longer true as screening labs are using second tier GAG testing from DBS to reduce MPS I positives. It is important to point this out as this would in theory decrease the need for variant analysis and follow-up diagnosis.  The manuscript is: “Precision newborn screening for lysosomal disorders.” Baerg MM et al,  Genet Med. 2018 Aug;20(8):847-854. doi: 10.1038/gim.2017.194. Epub 2017 Nov 9.  This would seem to be preferred over performing sequencing as variant analysis is often not conclusive nor something that can always be done quickly.
  2. Reference 13 seems to be incomplete.
  3. In introduction, it is mentioned that p.Ser633Leu is a variant associated with Scheie, I think it would be important to point out that Scheie is a mild form of MPS I, and that this variant is good for comparing to variants producing pseudo-deficient enzyme.
  4. Table 1 show p.Glu182Lys variant as pathogenic, and is described as a catalytically dead variant, I presume this is associated with the severe MPS I phenotype, can you clarify?
  5. Do you think the platform you used here could be used with other LSD enzymes such as GAA and GALC which are part of newborn screening across multiple states?

Author Response

  1. This is an excellent point raised by the reviewer and we have modified the introduction accordingly.  We now discuss the benefits of second-tier GAG testing in DBS and have included the reference provided by the reviewer.
  2. This was one of several typos generated when the managing editor of the journal reformatted our manuscript into the journal's template.  We regret that we overlooked these errors upon final review but have fixed all the typos, font issues and other nomenclature mistakes.
  3. We agree with the reviewer and have included more detail about the distinction between Scheie and Hurler.
  4. This is another good point.  The answer is yes.  The only patient reported (that we are aware of) with the E182K variant also has the W402X change.  This presents with severe Hurler as expected.  We made a note of this in the revised manuscript.
  5. Yes - in theory this type of analysis is applicable to other LSDs.  Our efforts are expanding into MPSII at the moment but other LSDs for which NBS is available would benefit from this general platform if second-tier testing is not possible.

Reviewer 2 Report

The manuscript titled "A biochemical platform to define the relative specific activity of IDUA variants identified by NBS" by Yu et al is an interesting and novel concept to do functional studies for novel VUS identified in MPSI. 

As authors have rightly put it, due to inclusion of MPSI disorder on RUSP NBS panel and extensive MPSI sequencing efforts, many novel variants are being identified in NBS positive patients with no information available about their pathogenicity.  As a result, the list of VUS keeps growing and clinicians and families with positive NBS baby have to go through many years of uncertainty and tough decisions regarding patient management.  Development of a robust and reliable functional assay platform to categorize these VUS as pathogenicity/disease causing or benign variants is an urgent unmet need in medical community.  Having said that - the functional analysis platform presented in this paper is novel and interesting, but it is not ready for implementation based on the data and experiments authors have presented.

-Throughout the manuscript there are font/font size and color changes - which is very distracting and reflects bad and unprofessional when reading the manuscript.  It seems that paragraphs and text has been cut and pasted without paying attention to how they look and fit in.  This problem exists in every section of manuscript, please edit and have uniformity in the manuscript.

Introduction:

- line 42-43 does not make sense.

-Line 46-47 does not make sense, why would clinicians be removed from diagnostic algorithm?  Follow up team does/must have clinicians involved in follow up testing and treatment.

-Line 54 - Pompe disease, another LSD testing on RUSP panel does have a huge Pseudo-deficiency allele and VUS identification problem as well.  Authors have failed to mention that, as that is the only other LSD on RUSP panel currently.

-Line 55-56 -does not make sense, please clarify.

Line 58-59 - GAGs can also be measured on DBS card (published already) as a second tier diagnostic test, urine sample is not the only sample used for GAG measurement.  Good idea for authors to expand on 'how good is the correlation between the IDUA enzyme, IDUA mutation testing and GAG measurement in DBS data with clinical symptoms for MPSI (published already)

Material and Methods: 

-Why did authors decide to pick 2 missense pathogenic variants with variable enzyme activity levels (attenuated) as positive control for expression studies and not pick/include any of the severe Hurler variants (nonsense or FS del/dup variants)?

-line 104 - 3ul or ~ 11ul of lentivirus what does ~ sign mean here and the lentiviral particle units in parenthesis ~ ?

-lines 104-108, can be rewritten clearly, a bit confusing to make sense of it.  What is the Puromycin concentration used - 1.0 or 0.1ug/ml.

-DPBS - need to write full word for this abbreviated acronym.

-Ex/Em - write full words for these abbreviations.

-Relative enzyme activity measurement concept is interesting - more details with how they came about it and table showing densitometry readings from Western blots for each clone and IDUA enzyme activity measured for each clone and then calculated relative activity would be very helpful to the reader.

Results-

-Why only 2 individual clones were analyzed for enzyme activity and Western blot analysis, may have been better to have 4 or more replicates due to variability induced by the viral integration and in vitro expression system and variability seen between the 2 clones analyzed.

-Lines 190-192 - Protein abundance concept is confusing - Western blots also have a known amount of protein loaded onto it.

-It is important to mark the size of the mature and pre-processed protein in Western blot gels. Important to state what is the size of the active protein form and what are various pre-processed protein forms during protein trafficking through ER - Golgi complex.  Authors can not make a statement about difference in enzyme/protein processing without commenting on how this protein gets processed.  All Western blot gels need molecular weight markers on it.

-Line 203 - needs editing/correction. 

-Line 218 - Is it and between the 2 variants listed or OR.

-Authors need to explain the huge variability they have seen within the same variant, they should have done more clone replicated if this variability was observed.  Cannot conclude results based on 2 variable measurements.

-Need more details of this normalization process used to calculate relative abundance IDUA activity for VUS before it is claimed to be a sensitive system as its strengths.

Line 261 - very confusing and bold statement - please clarify and explain more.

-Line 277 -What is p. Ale75 Thr.  Line 289 - p.Ale79Thr - what is this?  Needs editing and correction.

-Line 287-289-and line 292=293-  What are these statements - does not make sense?  Confusing..

-Why are authors are using 2 missense variable activity variants as a gold standard comparison for a variant to define it as attenuated or severe variant.

- If authors were aware of the literature and knew that HAP1 cell lines have very poor transduction efficiency and lentiviral particles are cumbersome and hard to produce to create variant constructs, then why did they decide on using this cell line and lentivirus system?  

Author Response

-Throughout the manuscript there are font/font size and color changes - which is very distracting and reflects bad and unprofessional when reading the manuscript.  It seems that paragraphs and text has been cut and pasted without paying attention to how they look and fit in.  This problem exists in every section of manuscript, please edit and have uniformity in the manuscript.

We thank the reviewer for their careful and thorough reading of the manuscript and have made every attempt to address their concerns. The font and style changes occurred when the managing editor put the manuscript into the IJNS template and then converted it to a PDF. We deeply regret that we did not detect these formatting issues earlier upon our final review. We have corrected these issues in our revision.

Introduction:

- line 42-43 does not make sense.

There was a period missing in this sentence that we corrected.

-Line 46-47 does not make sense, why would clinicians be removed from diagnostic algorithm?  Follow up team does/must have clinicians involved in follow up testing and treatment.

We agree with the reviewer that this statement is misleading. The intent is not to remove the clinician from the diagnostic algorithm but rather to convey that the diagnostic capability is limited when no clinical features are present in the proband. We modified this sentence to better reflect that concept.

-Line 54 - Pompe disease, another LSD testing on RUSP panel does have a huge Pseudo-deficiency allele and VUS identification problem as well.  Authors have failed to mention that, as that is the only other LSD on RUSP panel currently.

We agree Pompe should have been mentioned and have now included it in this section.

-Line 55-56 -does not make sense, please clarify.

We restructured this entire paragraph to more simply indicate that pseudodeficiency is problematic for newborn screening as it creates increased false positives in the near term until those alleles are identified and characterized as pseudodeficiency variants.

Line 58-59 - GAGs can also be measured on DBS card (published already) as a second tier diagnostic test, urine sample is not the only sample used for GAG measurement.  Good idea for authors to expand on 'how good is the correlation between the IDUA enzyme, IDUA mutation testing and GAG measurement in DBS data with clinical symptoms for MPSI (published already)

This is now mentioned in the introduction and we discuss the recently published work on GAG measurements in DBS. We also added two new references (Herbst et al and Peck et al) that address this issue and provide examples in their work where the correlation is still not perfect between GAG measurements and enzyme activity in DBS.

Material and Methods: 

-Why did authors decide to pick 2 missense pathogenic variants with variable enzyme activity levels (attenuated) as positive control for expression studies and not pick/include any of the severe Hurler variants (nonsense or FS del/dup variants)?

We chose two missense variants associated with attenuated disease (p.His240Arg and p.Ser633Leu) as we felt these would be useful to establish the “Scheie” range for our platform. For Hurler, we chose as the p.Glu182Lys variant as it was straightforward to generate by site-directed mutagenesis prior to lentiviral production, and because it essentially produces a catalytically dead enzyme which makes for a good comparison to the activity in the KO cell line. As we expand the platform and the number of variants tested, we anticipate even better granularity within the Hurler and Scheie ranges.

-line 104 - 3ul or ~ 11ul of lentivirus what does ~ sign mean here and the lentiviral particle units in parenthesis ~ ?

We have corrected this and clarified this sentence. Two different amounts of virus were utilized based on two different preparations.

-lines 104-108, can be rewritten clearly, a bit confusing to make sense of it.  What is the Puromycin concentration used - 1.0 or 0.1ug/ml.

The concentration of puromycin was switched from a higher one to ten-fold less following the original selection step. In other words, the 0.1 µg/mL is the concentration used for maintenance of selected cells and clones. We added some text on the use of different concentrations for the benefit of the reader.

-DPBS - need to write full word for this abbreviated acronym.

DPBS is now spelled out as “Dulbecco’s Phosphate-Buffered Saline”.

-Ex/Em - write full words for these abbreviations.

Excitation/Emission is spelled out now.

-Relative enzyme activity measurement concept is interesting - more details with how they came about it and table showing densitometry readings from Western blots for each clone and IDUA enzyme activity measured for each clone and then calculated relative activity would be very helpful to the reader.

We thank the reviewer for this comment and agree that more detail is warranted on how the relative specific activity is calculated. While the activity determinations are straightforward, the assessment of relative enzyme abundance is not as obvious. We made a few changes to the Methods section to clarify our process, including more detail on the use of a standard range of WT lysate in order to more accurately determine relative enzyme abundance. We also modified Figures 3 and 4 to now show the relative iduronidase abundance values for the different clones. The final relative specific activities are calculated by dividing the activity values by these relative abundance values. We believe these changes adequately address the reviewer’s comment.

Results-

-Why only 2 individual clones were analyzed for enzyme activity and Western blot analysis, may have been better to have 4 or more replicates due to variability induced by the viral integration and in vitro expression system and variability seen between the 2 clones analyzed.

In establishing the platform, we debated how many clones and replicate analyses in each clone were needed to properly stratify the variants with regard to relative specific activity. Based on our analysis, we felt that the highly reproducible specific activities we obtained from two different clones (with different expression levels) were sufficient, especially for the pseudodeficiency variants we studied. For the two Scheie-associated variants, we performed triplicate analysis on the two different clones to confirm the differences between them but, in general, we are of the opinion that analyzing four different clones is likely not needed unless there is clear discrepancy between the values obtained for the first two clones studied.

-Lines 190-192 - Protein abundance concept is confusing - Western blots also have a known amount of protein loaded onto it.

Gels were loaded with the same amount of total lysate protein but we can only estimate the actual amount of iduronidase enzyme that is present. This estimation is done relative to the pixel density for the enzyme on the blot, and then compared to the range of iduronidase enzyme detected in WT lysate. In other words, we can’t formally express the signal detected as a protein concentration, rather only as a relative abundance. We think our expanded clarification of how relative specific activity is determined will be helpful for the reader.

-It is important to mark the size of the mature and pre-processed protein in Western blot gels. Important to state what is the size of the active protein form and what are various pre-processed protein forms during protein trafficking through ER - Golgi complex.  Authors cannot make a statement about difference in enzyme/protein processing without commenting on how this protein gets processed.  All Western blot gels need molecular weight markers on it.

We agree with the reviewer and added such markers to the blots. We were surprised at how little processing was observed on the overexpressed enzyme in HAP1 cells (compared to CHO cells, for example). The use of the HAP1 cell system may therefore not be optimal in the accurate assessment of variants that impact processing. In comparing our results with the Western blot data with the CHO cell results from Matte et al., we did feel that some of the processing effects were consistent, such as the observation by both groups that E182K enzyme shows a similar profile of processed forms as the WT enzyme.

-Line 203 - needs editing/correction. 

We edited this line to better clarify our observation. Essentially, differences in expression among the various clones may relate to where the virally delivered sequence integrated into the genome.

-Line 218 - Is it and between the 2 variants listed or OR.

We believe we fixed the correct inquiry on line 216. It should be “and” not “or” in that place.

-Authors need to explain the huge variability they have seen within the same variant, they should have done more clone replicated if this variability was observed.  Cannot conclude results based on 2 variable measurements.

We are unclear where the reviewer has noted huge variability in our analysis of different clones. In fact, we were encouraged to see such reproducible values for relative specific activity despite the difference in expression between the clones. In other words, even though the expression of the enzyme is variable between clones, the relative specific activity that is calculated is highly similar. Based on this reproducibility, we did not feel that it was necessary to analyze more clones. For the two Scheie variants, however, we did perform triplicate analysis of the two clones we chose for each to gauge the variability and confirm the 5-fold difference we initially detected. The results we obtained were highly reproducible. Ultimately, we believe the analysis of three or more individual clones may be necessary for some variants that give ambiguous results but is excessive for most variants, especially when the initial determination clearly supports benign or pseudodeficiency status.

-Need more details of this normalization process used to calculate relative abundance IDUA activity for VUS before it is claimed to be a sensitive system as its strengths.

As stated above, we included more information on how the relative specific activity was calculated and modified the figures to show the relative abundance values obtained by densitometry. We also tried to clarify what is intended when we say sensitivity. This refers to the fact that variant enzymes with low activity can still be reliably measured because they are overexpressed in this cell-based system.

Line 261 - very confusing and bold statement - please clarify and explain more.

The reference of specific line numbers by the reviewer does not appear to match the statements in question so we hope we are addressing the right issues. If the intent here is to clarify the statement about sensitivity, we believe we have now done so in the revised manuscript and make it clear that sensitivity is beneficial for variants that produce enzyme with very low residual activity.

-Line 277 -What is p. Ale75 Thr.  Line 289 - p.Ale79Thr - what is this?  Needs editing and correction.

These were unfortunate typos created during the transfer of our manuscript to the IJNS template. They have been fixed.

-Line 287-289-and line 292=293- What are these statements - does not make sense?  Confusing.

This section of the discussion has been modified to improve clarity.

-Why are authors are using 2 missense variable activity variants as a gold standard comparison for a variant to define it as attenuated or severe variant.

We are uncertain we fully understand this comment but believe our choice of variants used for the optimization of this platform are adequately justified above in response to a related comment by the reviewer.

- If authors were aware of the literature and knew that HAP1 cell lines have very poor transduction efficiency and lentiviral particles are cumbersome and hard to produce to create variant constructs, then why did they decide on using this cell line and lentivirus system?  

This is a fair point. To explain, we were able to obtain the IDUA HAP1 KO cells quickly and thought initially that transduction would be the easiest way to generate the single cell clones. We did not realize the transduction efficiency would be as poor as it was but decided to continue with our viral-based platform in light of the investment made on the lentiviral particles.

Reviewer 3 Report

The paper by you et al presents novel IDUA variants identified through newborn screening by applying the highly sensitive biochemical platform. It can be used to test variant pathogenicity and confirm pseudodeficiency.

The strength of this platform is its sensitivity.

Specific activity estimations at the lower end of the range are important to distinguish Scheie and Hurler-associated variants, especially when they are combined with other unknown variants. It automatically affect clinical decisions regarding the management of these cases.

The actual expression level is an important factor to consider and it is possible for patients with same mutation can show different phenotype.

The report is well-written and presents results graphically which is easy to follow. 

Suggestion: please use the new nomenclature of mutations. The authors use p.Trp402* and then W402X. The same with H240R  and p.His240Arg.

Author Response

We corrected the oversight of standard nomenclature throughout the manuscript.

Reviewer 4 Report

two minor comments:

Please use always the same nomenclature to identify DNA variants

Ref 12 and 13 are the same

Author Response

We went through the manuscript again and fixed all the nomenclature errors throughout, even in the figures where we felt the use of the single letter coding was preferred to avoid excessive clutter.

We also fixed the reference issue.